# Irisin Attenuates Neuroinflammation Targeting the NLRP3 Inflammasome

**DOI:** 10.3390/molecules29235623

**Published:** 2024-11-28

**Authors:** Francesca Martina Filannino, Melania Ruggiero, Maria Antonietta Panaro, Dario Domenico Lofrumento, Teresa Trotta, Tarek Benameur, Antonia Cianciulli, Rosa Calvello, Federico Zoila, Chiara Porro

**Affiliations:** 1Department of Clinical and Experimental Medicine, University of Foggia, I-71100 Foggia, Italy; francesca.filannino@unifg.it (F.M.F.); teresa.trotta@unifg.it (T.T.); federico.zoila@unifg.it (F.Z.); 2Department of Biosciences, Biotechnologies and Environment, University of Bari, I-70125 Bari, Italy; melania.ruggiero@uniba.it (M.R.); mariaantonietta.panaro@uniba.it (M.A.P.); antonia.cianciulli@uniba.it (A.C.); rosa.calvello@uniba.it (R.C.); 3Department of Biological and Environmental Sciences and Technologies, Section of Human Anatomy, University of Salento, I-73100 Lecce, Italy; dario.lofrumento@unisalento.it; 4Department of Biomedical Sciences, College of Medicine, King Faisal University, Al-Ahsa 31982, Saudi Arabia

**Keywords:** microglia, irisin, inflammasome, neuroinflammation

## Abstract

Neuroinflammation is defined as an immune response involving various cell types, particularly microglia, which monitor the neuroimmune axis. Microglia activate in two distinct ways: M1, which is pro-inflammatory and capable of inducing phagocytosis and releasing pro-inflammatory factors, and M2, which has anti-inflammatory properties. Inflammasomes are large protein complexes that form in response to internal danger signals, activating caspase-1 and leading to the release of pro-inflammatory cytokines such as interleukin 1β. Irisin, a peptide primarily released by muscles during exercise, was examined for its effects on BV2 microglial cells in vitro. Even at low concentrations, irisin was observed to influence the NLRP3 inflammasome, showing potential as a neuroprotective and anti-inflammatory agent after stimulation with lipopolysaccharides (LPSs). Irisin helped maintain microglia in their typical physiological state and reduced their migratory capacity. Irisin also increased Arg-1 protein expression, a marker of M2 polarization, while downregulating NLRP3, Pycard, caspase-1, IL-1β, and CD14. The results of this study indicate that irisin may serve as a crucial mediator of neuroprotection, thus representing an innovative tool for the prevention of neurodegenerative diseases.

## 1. Introduction

Microglia are a resident macrophage cell type of the central nervous system (CNS), derived from myeloid cells. They are the primary immune effector cells in the brain, constituting the initial line of defense against damage or injury of the brain. Microglia are sensitive to minor disturbances in central nervous system homeostasis and are capable of undergoing morphological and functional transformations during activation [1,2]. In a healthy brain, resident microglia are thought to be in a resting state. However, upon activation, they undergo structural changes, including the development of motile branches or the migration of the soma [3,4,5]. Microglial activation is triggered in response to various forms of brain injuries, which, in turn, initiate inflammatory responses. Such injuries include head trauma, ischemia, neurodegenerative diseases, autoimmune disorders, prion diseases, infectious diseases, and brain tumors [6,7]. The hallmark of microglial activation is their proliferation at the site of injury, as well as their recruitment and expression of immune response-associated proteins. These proteins include cytokines, like GF-β1, TNF-α, IL-1, and IL-6. Additionally, in vitro studies have demonstrated that cytokines including interferon-γ, IL- 1, IL-4, and TNF-α also induce comparable changes in microglia.

In 2012, irisin was identified as an exercise-induced myokine by Boström’s group [8]. It was initially recognized as a hormone released into the bloodstream following the proteolytic cleavage of fibronectin type III domain-containing protein 5 (FNDC5) in skeletal muscles [9,10].

FNDC5 is composed of an N-terminal signal sequence, a fibronectin type III (FNIII) domain, an unidentified domain, a hydrophobic transmembrane domain, and a cytoplasmic C-terminus. The N-terminal signal peptide serves as an endoplasmic reticulum trafficking signal, which is essential for the maturation and cleavage of FNDC5 [8,11,12]. Interestingly, irisin is also expressed in human and mouse brain and cerebrospinal fluid (CSF) [13]. Notably, peripheral irisin can cross the blood–brain barrier [14,15], and spread-out in the CSF, where it mediates exercise-induced beneficial effects on cognition [16].

Neurodegenerative diseases have been linked to several inflammasomes, with the pyrin domain-containing 3 (NLRP3) inflammasome being particularly important for the development and progression of these conditions [17]. The inflammasome is a multimeric protein complex, consisting of a sensor, an adaptor, and the downstream effector caspase-1 [18]. The activation of the NLRP3 inflammasome involves interactions between molecules containing an N-terminal pyrin domain (PYD) and a C-terminal caspase activation and recruitment domain (CARD). These interactions lead to the formation of a ring-like perinuclear complex known as the “ASC speck”, a hallmark of canonical inflammasome activation [19,20,21,22]. Upon inflammasome activation, the adaptor protein ASC interacts with the CARD of caspase-1, recruiting procaspase-1 [23,24]. Procaspase-1 is then converted into its bioactive form by proximity-induced autocatalysis, generating mature caspase-1, which cleaves pro-IL-1β and pro-IL-18 into their respective secreted forms [25,26]. Caspase-1 also mediates the cleavage of pore-forming gasdermin D (GSDMD), which induces pyroptosis, a lytic and pro-inflammatory form of cell death [27,28]. Previous studies have suggested that the specific abnormal deposition of specific proteins may be a trigger for NLRP3 inflammasome activation in neurodegenerative diseases. In Alzheimer’s disease (AD), the accumulation of the amyloid-β (Aβ) protein may promote the assembly and activation of the NLRP3 inflammasome, leading to subsequent inflammatory events [29]. For these reasons, inhibiting this inflammasome has been shown to exert neuroprotective effects [17,30,31]. Therefore, emerging strategies for combating neurodegenerative diseases may include treatments that target both upstream and downstream pathways of the NLRP3 inflammasome [30,32,33].

Here, we discuss the impact of irisin on the NLRP3 inflammasome pathway, as well as the potential role of this pathway in Parkinson’s disease (PD), AD, and amyotrophic lateral sclerosis (ALS), and the therapies that target the NLRP3 inflammasome [34,35,36].

Previous studies showed that irisin alleviated inflammatory responses and microglial activation induced by LPS by inhibiting the NF-κB/MAPK/IRF3 signaling pathways in BV-2 microglial cells [37,38,39,40], suggesting its potential neuroprotective effects [41,42,43].

The present study aims to evaluate the in vitro biological effects of irisin on the BV2 microglial cellular model in culture, focusing on its potential as a neuroprotective and anti-inflammatory agent. By assessing its effects on the NLRP3 inflammasome pathway, this research seeks to demonstrate irisin’s ability to modulate various inflammatory and anti-inflammatory pathways involved in the pathogenesis and progression of neurodegenerative diseases.

## 2. Results

### 2.1. Influence of Irisin on BV2 Cells

As shown in (Figure 1A), no cytotoxic effects were detected at all tested concentrations (5–20 nM) of irisin. The range of samples used was selected on the basis of previous studies on brain cells incubated with irisin [44,45]; at concentrations starting of 5 nM, irisin was able to stimulate proliferation. Therefore, the MTT assay was repeated at 24 h with an irisin concentration of 5 nM in the presence or absence of 1 µg/mL LPSs (Figure 1B). Pretreatment with irisin and LPSs for 24 h did not significantly affect the viability of BV-2 microglial cells compared to the control.

### 2.2. Irisin Induces Variability in the Morphology of BV2 Cells

Microglial cells are nervous system-specific immune cells that serve as tissue-resident macrophages [46,47]. They are able to polarize into different phenotypes and retain the ability to switch functions to maintain tissue homeostasis [48,49]. The morphological development of microglial cells is characterized by the progressive development of processes and cellular soma. Figure 2A shows that untreated cells, corresponding to the control condition, exhibit a typical microglial morphology in the quiescent state [50,51,52]. This includes a small central body and numerous elongated processes. As expected, treatment with LPSs (1 µg/mL) (Figure 2B) [53,54,55] resulted in BV2 cells acquiring an amoeboid morphology, which is linked to a pro-inflammatory condition. This was characterized by an increase in their soma and a reduction in their prolongation, highlighting an amoeboid morphology [50,51,52,53,54,55,56]. Treatment with irisin (Figure 2C) did not affect cell morphology with respect to the control. The cells exhibited a branched morphology with a small central soma, which is associated with the resting state [57].

Similarly, in cells treated with irisin and LPSs (Figure 2D), irisin was observed to reverse the typical amoeboid phenotype induced by LPSs, resulting in greater branching of the distal branches. The analysis of cellular areas (Figure 2E) confirmed the results, revealing that LPS treatment increased the size of BV2 cells, characteristic of the amoeboid form. The cellular area values were significantly higher compared to the control condition. However, when the cells were treated with irisin in the presence of LPSs (1 µg/mL), irisin significantly reduced the LPS-induced increase in cellular area.

### 2.3. Effect of Irisin on Reducing Cell Migration

A wound closure assay was used to assess the effect on microglial motility after irisin treatment. For the assay, a scratch was made with a scraper, and BV2 cells were allowed to migrate within the scratch during a 24 h incubation period. The fixed and free cell areas were then measured to determine whether irisin stimulation, in the presence or absence of LPSs, resulted in wound closure [58,59,60,61]. The results of this experiment are shown in Figure 3. The data indicate that stimulation with LPSs resulted in a significantly greater potential for cell migration compared to the control cells after 24 h of incubation in the free wound area after excision (Figure 3C). After the application of irisin (Figure 3D), as shown in Figure 3, there was no significant increase in BV2 cell motility, with a free wound surface which was almost comparable to the control condition after 24 h (Figure 3F). This indicates that the migratory capacity of these cells was significantly lower than in LPS-stimulated cells. Figure 3F demonstrates that, in cells treated with both LPSs and irisin, irisin reversed the effect of LPSs, leading to a significant reduction in cell migration compared to BV2 cells treated with LPSs alone. Thus, in both migration assays, irisin worked as a regulator of migratory capacity by reducing the LPS-induced increase in BV2 cell motility.

### 2.4. NLRP3 Inflammasome Activation Decreased in BV2 Cells Treated with Irisin

The Western blot assay and semi-quantitative analysis showed that NLRP3 expression was significantly reduced in BV2 cells treated with irisin compared to cells treated with LPSs. Similarly, the expression of Pycard significantly decreased with the use of irisin. Moreover, we could see a clear reduction in the expression of these two pro-inflammatory factors in BV2 cells treated with both irisin and LPSs with respect to cells treated with LPSs alone. Taken together, these data demonstrate that the NLRP3 inflammasome is reduced in microglial cells under irisin conditions (Figure 4).

### 2.5. Irisin’s Effects on Pro-Inflammatory Marker Expression

We further investigated the role of irisin in inhibiting LPS-induced inflammation in the microglia. Irisin treatment downregulated NLRP3-induced inflammasome activation, as reflected by a reduction in the protein expression levels of IL1β (Figure 5A) and caspase-1 (Figure 5B) in cells pretreated with irisin and LPS compared to those treated with LPS alone over 24 h. We then examined the expression of CD14 (Figure 5C), a glycosylphosphatidylinositol-anchored monocyte antigen, which, together with TLR4, represents an early event in the activation of the neuroinflammatory signaling pathway. Upon binding to LPS, CD14 associates with the extracellular domain of TLR4 [62,63]. Again, the protein expression level of CD14 was upregulated in response to LPS exposure and was reduced by irisin pretreatment in LPS-treated cells.

### 2.6. Irisin’s Effects on Anti-Inflammatory Marker Expression

To further investigate the effect of irisin on the phenotypic transition of BV2 cells, Arg-1 was assessed as a marker of M2 microglial polarization [64]. The results demonstrated that irisin alone significantly increased Arg-1 levels compared to the control. In contrast, LPS treatment reduced Arginase 1 (Arg-1) expression relative to the control. Notably, co-treatment with irisin and LPSs produced the opposite effect, resulting in elevated Arg-1 expression in BV2 cells (Figure 6).

## 3. Discussion

In the development and maintenance of neuronal homeostasis, resting microglia are actively implicated [65,66,67]. In the resting state, microglia scan the microenvironment in real time with their ramified processes contributing to tissue homeostasis [68]. Microglia can be activated by a variety of substances, including foreign pathogens, abnormal protein aggregates, and apoptotic cells [68,69]. Several stimuli have been shown to induce classical microglial activation. These include LPS, interferon-γ, and in vivo β-amyloid and α-synuclein [68,70,71]. Activation is characterized by a series of cellular and molecular events leading to the production of a wide variety of inflammatory mediators and cytotoxic molecules, and a drastic change from a highly branched morphology to an amoeboid form [72,73]. Normally, these immune responses are closely controlled in microglia to maintain tissue homeostasis. Traditionally, depending on the lesion and stimulus inducing activation, microglia can adapt to polarized phenotypes of classical activation, such as the pro-inflammatory M1 phenotype, or alternatively acquired activation/deactivation, that is, the immunosuppressive M2 phenotype [70,74,75]. When activated to the M1 phenotype in response to an insult, microglia secrete high levels of pro-inflammatory factors, including IL1-1β, IL-6, and TNFα, along with increased production of NO and ROS. They are the first line of defense, as they can eliminate pathogens. The excessive levels of these factors that result from chronic microglial activation; however, they can also cause neuronal damage: this has been confirmed for several diseases such as AD, PD, and ALS [76,77,78].

LPS is a prominent heat-stable glycolipid component of the outer membrane of Gram-negative bacteria that induces a strong inflammatory response [79,80,81,82]. Consequently, microglia carrying activated TLR4 produce inflammatory mediators. LPS recognizes and binds to the LPS-binding protein and the glycosylphosphatidylinositol-anchored protein CD14, which further associates from TLR4 and activates pro-inflammatory signaling cascades to induce the production and release of TNFα, IL-6, and NO, leading to neuronal cell death [83,84].

Within damaged tissue, microglia exist in different states of activation and retain the ability to change their functional phenotype during the inflammatory response [46]. Microglial cells, in their fully ramified form, are present in the normal CNS. Here, they provide neurotrophic substances, act on and regulate neurotransmitters and hormones, and protect neurons from damage by responding to changes in the microenvironment [85]. In this role, microglial cell bodies can be elongated and nodular, rod-shaped, or tortuous with branching and bulging processes, or they can be more radial with pin-like figures [86]. The MTT assay was used to determine the effects of different concentrations of irisin on BV2 cells’ viability. The lowest concentration able to stimulate cell viability was 5 nM, so it was chosen for subsequent experimental tests. Pretreatment with irisin and LPS for 24 h did not significantly affect the viability of BV-2 microglial cells compared to the control, as already reported in some scientific studies [45].

In addition, microglia have migratory and phagocytic abilities that modulate the CNS microenvironment in response to homeostatic changes. These cells can undergo a morphological switch to a pro-inflammatory or anti-inflammatory phenotype, depending on the signals they receive [87]. The basic motility of microglia is characterized by the movement of typical branching cell processes that are part of the microglial morphology in the ‘resting state’. When adopting the pro-inflammatory phenotype (M1), microglia retract these branching processes and migrate towards areas of tissue damage or injury, displaying an amoeboid morphology. In contrast, during the alternative activation (M2) phenotype, the cells retain branching cell processes and a reduced cytoplasm, showing a reduced migratory capacity, similar to that of resting microglia [88,89]. Our results demonstrate that irisin-treated microglia exhibit a resting phenotype, unlike the amoeboid form induced by LPS stimulation.

The induction of the M1 phenotype by LPSs was confirmed by the increase in the expression of CD14, a marker of microglial activation. CD14, like many molecules related to immune function produced by microglia, is subject to regulation by other molecules present in the inflamed environment of the CNS. CD14 is a ligand for cell wall molecules of various pathogens or membrane structures of apoptotic cells. CD14 mediates the activation of monocytic cells by cytokines such as IL-2, hsp60, or hsp70, triggering the production and release of mediators such as TNFα, IL-1, IL-6, IL-8, IL-18, and IFNβ, as well as chemical mediators such as NO [85,86]. The CD14-mediated activation of three mitogen-activated protein kinase pathways (ERK1/2, JNK, and p53) leads to increased phagocytic activity [85,86,87,88,89,90]. In our study, we saw how LPS-induced inflammation could be reversed by administering a very small dose of irisin. In fact, the use of irisin induced a change in cell morphology in cells treated with irisin and LPS, reducing the presence of pro-inflammatory amoeboid M1 cells. At the same time, there was a significant reduction in the expression of CD14, confirming an anti-inflammatory effect. We verified that treatment with 5 nM Irisin could lead to an increase in Arg-1 protein expression as a marker of M2 polarization [64]. Irisin treatment increased this ratio towards Arg-1 and promoted microglial cell polarization towards the M2 phenotype. Many articles have shown the anti-inflammatory activities of irisin on the CNS. It is well known that physical exercise has a slowing effect on neurodegenerative processes. Regular exercise is associated with a reduced risk of dementia and other neurodegenerative diseases like AD and PD [91,92]. Physical activity reduces chronic oxidative stress mitochondrial biogenesis and enhances the positive regulation of autophagy in PD [90]. Additionally, exercise stimulates the synthesis of key neurotransmitters such as dopamine and various trophic factors, including the glial-derived neurotrophic factor, the insulin-like growth factor-1, the brain-derived neurotrophic factor, and the fibroblast growth factor-2 [93]. In this study, we aimed to further characterize the anti-inflammatory properties of irisin by investigating its possible influence on the NLRP3 inflammasome.

Kim H. et al., in 2018, showed that irisin binds to proteins of the αV class of integrins, and their biophysical studies identified interacting surfaces between irisin and αV/β5 integrin. In fact, the chemical inhibition of αV integrins by irisin blocks signaling and function in osteocytes and fat cells [94], while, in microglial cells, irisin binds to αV/β5 integrin [95], and, after binding to its receptor, irisin activates its pathway in cells.

Recently, a lot of attention has been paid to the role of the NLRP3 inflammasome in CNS diseases. Regarding multiple sclerosis, an inflammatory demyelinating disease of the CNS, several studies suggest a general involvement of the NLRP3 inflammasome. In addition to MS, NLRP3 inflammasome involvement in AD has been demonstrated both in vitro, by the ability of amyloid beta peptides to activate the inflammasome, and in vivo, in a mouse model of AD, where NLRP3 knockout mice were protected from spatial memory impairment and showed reduced Aβ plaque burden [41,96,97]. Similarly, in PD, the accumulation of protein aggregates such as Lewy bodies has been shown to induce inflammasome-dependent IL-1β secretion in the microglia, suggesting the involvement of NLRP3 [98]. They are required for the activation of the protease caspase-1 and the downstream secretion of two of its substrates, the pro-inflammatory cytokines IL-1β and IL-18 [99,100]. Inflammasomes are multi-protein complexes expressed mainly in myeloid cells and involved in the activation of the protease caspase-1 and the downstream secretion of the pro-inflammatory cytokines IL-1β and IL-18 [101,102]. To determine whether irisin inhibits NLRP3 inflammasome activation in the microglia, we examined the expression of NLRP3, Pycard, and caspase-1 inflammasome components in BV2 cells subjected to stimulation with irisin, LPS, and both.

The levels of the NLRP3 inflammasome components were increased in response to stimulation with LPS; in contrast, irisin lowered the expression of NLRP3, Pycard, and caspase-1, suggesting its inhibitory activity on the NLRP3 pathway, thus playing an important role in mediating inflammation and subsequent cell death [23].

Furthermore, the modulation of the expression of IL-1beta in BV2 cells treated with irisin and LPS confirmed the anti-inflammatory effect of irisin on this pathway. IL-1beta plays an important role in the stimulation of the innate immune system and neuroinflammatory disease processes. As shown in a previous study, irisin can inhibit IL-1beta expression by downregulating NF-κB [102,103,104]. In this study, we observed that irisin, at low concentrations in the physiological range, was also able to inhibit the canonical NLRP3 inflammasome pathway and, consequently, IL-1beta release [105].

In recent years, the link between physical activity and the prevention of neurodegenerative diseases has attracted research attention for its potential applications in the medical field, particularly in the treatment of metabolic diseases such as diabetes and neurodegenerative diseases. The aim of this study was to demonstrate that irisin, a molecule released during exercise, is able to regulate microglial inflammation.

Our in vitro findings demonstrate that, even low levels, such as the physiological levels present in the central nervous system, irisin is capable of reducing microglial inflammation, a critical factor in the progression of neurodegenerative diseases. In particular, this study demonstrates that irisin exerts a protective function, downregulating the pro-inflammatory amoeboid phenotype induced by LPS. The attenuation of the inflammation induced by irisin was confirmed by the reduction in cell migration and the decrease in the expression of the pro-inflammatory marker CD14. Moreover, previous research indicated that irisin possessed anti-inflammatory properties. In this study, we further examined the influence of irisin on the NLRP3 inflammasome pathway. It was observed that irisin was capable of reducing inflammation by negatively regulating the expression of NLRP3, caspase 1, Pycard, and IL-1beta. The latter interleukin plays a pivotal role in maintaining and prolonging neuroinflammation.

In conclusion, our findings suggest that irisin administration may represent an efficacious therapeutic strategy for modulating inflammatory responses, notably in neurodegenerative contexts. Nevertheless, further research is necessary to fully elucidate the mechanisms by which irisin exerts its effects and evaluate its efficacy and safety.

## 4. Materials and Methods

### 4.1. Microglial Cell Culture

In this study, the BV2 murine microglial cell line, purchased from the American Type Culture Collection (Manassas, VA, USA), was used. The cells were cultured in Dulbecco’s Modified Eagle Medium (DMEM, Euroclone, Milan, Italy) supplemented with 10% fetal bovine serum (FBS; Euroclone, Milan, Italy), 100 U/mL penicillin, 100 μg/mL streptomycin, (penicillin–streptomycin; Euroclone, Milan, Italy), and 2 mM glutamine (glutamine; Euroclone, Milan, Italy) and maintained at 37 °C in a humidified incubator with 5% CO_2_. Adherent BV2 cells were detached with Trypsin-EDTA (Trypsin-EDTA 1X in PBS; Euroclone, Milan, Italy) then plated at an optimal density. After 24 h, one group of cells was treated with 5 nM irisin, while another with 1 μg/mL LPSs from *Escherichia coli* O128: B12 (Sigma-Aldrich, St. Louis, MI, USA). A third group of cells was pre-incubated with irisin and LPSs for 1 h. The cells were grown and collected after 24 h.

### 4.2. Irisin Solution Preparation

Irisin (Purity ≥ 95% (HPLC); Phoenix Pharmaceuticals Inc., Burlingame, CA, USA; CAS: 067-16) was initially prepared by dissolving it in DMSO (dimethyl sulfoxide, cell culture reagent; MP Biomedicals) to achieve a 1 M concentration. For the subsequent experiments, irisin was diluted in DMEM to prepare different dilutions from the stock solution. To investigate the in vitro effects of irisin, the cells were pre-incubated with irisin diluted in DMEM for a maximum of 24 h.

### 4.3. Cell Viability Assay

The cytotoxic effects of irisin on BV2 cells was assessed using an MTT assay, with Thiazolyl Blue Tetrazolium Bromide (Sigma-Aldrich (CAS: 298-93-1)). The BV2 cells were seeded in 24-well plates at a density of 2 × 10^5^ cells and maintained at 37 °C with 5% CO_2_ for 24 h. The initial irisin concentrations used to assess cytotoxicity were 5 nM, 10 nM, and 20 nM. Based on the results, the lowest effective concentration was selected to further investigate the potential protective effects of irisin. The MTT assay was then conducted using 5 nM of irisin, both in the absence and presence of 1 µg/mL LPSs, for 24 h. Absorbance was measured using a spectrophotometer (Filter Max F5 Multi-Mode Microplate Reader, Molecular Devices, San Jose, CA, USA) set to a wavelength of 595 nm. The results are expressed as a percentage (%) of cell viability relative to the control condition.

### 4.4. Cell Morphology Analysis

Microglial morphology was analyzed by measuring the cell body area to assess the effect of irisin on BV2 cells’ activation, both in the absence and presence of the pro-inflammatory stimulus LPS (1 µg/mL). About 5 × 10^5^ cells were plated in six-well plates. All morphological analyses were performed in triplicate. The results are expressed as the mean cell body area from five independent experiments, each measuring three cells per sample.

Cell morphology was examined using a Leica Microscopy photographic system (DM IRB Leica Microsystems GmbH, Wetzlar, Germany) at 10× and 20× magnifications. The ImageJ software was used to quantify cell areas (µm^2^).

### 4.5. Wound Healing Assay

To evaluate two-dimensional cell migration, we performed a wound healing assay. BV2 cells (1 × 10^6^) were seeded in a 6-well plate and grown to confluence. A monolayer was wounded with a sterile scraper. After washing with PBS and changing the DMEM, the remaining cells were incubated for 24 h under different conditions: with irisin only or in the presence of LPSs (1 µg/mL). All migration assays were performed in triplicate.

After 24 h, three pictures of each area of the wound were taken with photomicrographs of the different analyzed conditions. Wound closures/repairs were quantified using the ImageJ software and expressed as a percentage of the area covered by the cells relative to the initial wound size at the start of the assay.

### 4.6. Protein Extraction and Western Blot Analysis

Following the aforementioned treatments, the BV2 cells were detached from the plate by gentle scraping and collected after centrifugation at 2000 rpm for 10 min at 4 °C. The cells were then lysed using an ice-cold lysis buffer consisting of 50 nM Tris-HCl (pH8), 1% (*v*/*v*) Triton X-100, 1.5 M NaCl, 0.1% SDS, 1 μM leupeptin hemisulfate salt, 100 μM phenylmethylsulfonyl fluoride (PMSF), and 4 U/mL aprotinin (Sigma Aldrich).The lysates were collected by centrifugation at 12,000 rpm for 30 min at 4 °C after eight cycles of freezing and thawing.

A Western blot was performed as previously described [89], with the following primary antibodies: CD14 (1:500), IL-1β (1:500), caspase-1 (1:500), Arginase 1 (1:500), and β-actin (1:500) (Santa Cruz Biotechnology, Inc. Heidelberg, Germany). β-actin was used to normalize the protein expression levels [103]. Another set of primary antibodies was used; this included anti-NLRP3 (EPR23094-1, (1:1000)) and anti-TMS1/ASC (EPR10403, 1:1000) (Abcam). The membranes were incubated with the primary antibodies for 1 h at room temperature and then overnight at 4 °C before being incubated with horseradish peroxidase (HRP)-conjugated secondary antibodies (Santa Cruz Biotechnology, (1:10,000)) for 1 h at room temperature. Immunoreactive bands were detected using chemiluminescence (BioRad Laboratories, Hercules, CA, USA). The obtained bands were subjected to densitometric analysis using the ImageJ software. The results are expressed in arbitrary units.

### 4.7. Statistical Analysis

Statistical analyses were carried out using Statgraphics Centurion (Statgraphics Technologies Inc., The Plains, VA, USA). To analyze the data, we used two-sample comparison, one-way ANOVA, and Tukey’s post hoc tests. Data from five independent experiments are presented as means ± SDs, with all assays performed in triplicate. A *p*-value < 0.05 was considered statistically significant.

## Figures and Tables

**Figure 1 molecules-29-05623-f001:**
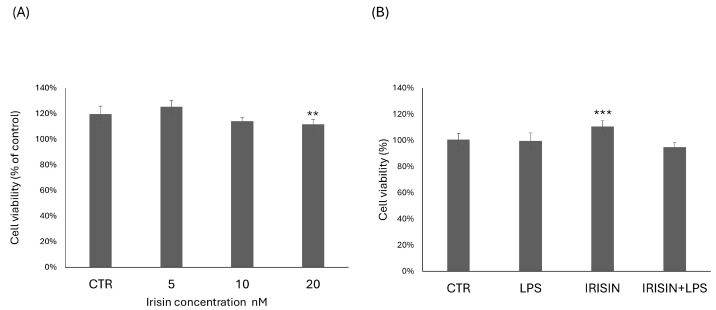
Effect of irisin on cell viability (MTT assay). BV-2 cells were incubated with irisin at concentrations spanning a dose–response curve, ranging from 5 nM to 20 nM (**A**). Irisin at a concentration of 5 nM was used to treat cells, either in the absence or presence of 1 µg/mL LPSs (**B**). Data are reported as percentages compared to control values and are expressed as means ± SDs. ** *p* < 0.01 compared to the control. *** *p* < 0.001 compared to the control.

**Figure 2 molecules-29-05623-f002:**
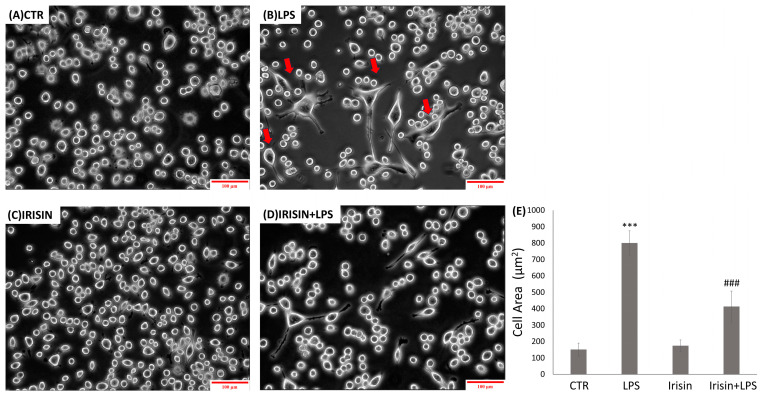
Morphological analysis of the BV2 cells following the administration of irisin, either alone or after LPS stimulation. The morphological analysis was conducted on BV2 cells in the control condition (**A**) and following the administration of 1 µg/mL LPSs (**B**), 5 nM irisin (**C**), or 5 nM irisin in the presence of 1 µg/mL LPSs (**D**). The scale bar is 100 µm (10× objective). The arrows in the images indicate cells that have undergone a morphological change. The cell areas (µm^2^) were quantified using the ImageJ 1.8.0 software, which was bound with Java 8 64-bit (**E**). The data are expressed as means ± standard deviations. A significant difference can be observed between the control and LPS groups (*** *p* < 0.001), as well as between the IRISIN + LPS and LPS groups (### *p* < 0.001).

**Figure 3 molecules-29-05623-f003:**
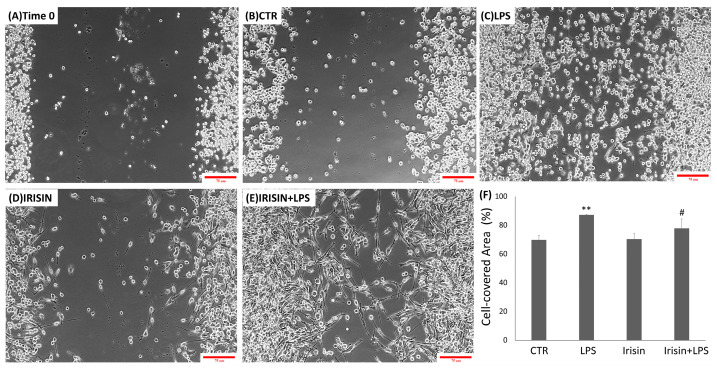
The analysis of the migratory capacity of microglia after the administration of irisin in the presence or absence of LPSs. A wound was generated in a sub-confluent layer of BV2 cells, and the resulting space was captured at the wound site 0 and 24 h after the treatment: (**A**) BV2 cells at time 0, (**B**) 24 h after the cut in the control condition, (**C**) treated with 1 µg/mL LPSs, (**D**) with 5 nM irisin, and (**E**) with 5 nM irisin in the presence of 1 µg/mL LPSs. The images are representative of an experiment with three independent replicates. The percentage of the wound gap was analyzed using the ImageJ software and subsequently plotted and statistically analyzed as the percentage of wound closure compared to the 0 time condition (**F**). The values are presented as means ± standard deviations. Bar: 75 µm (20× objective). ** *p* < 0.01 compared to the control. # *p* < 0.05 compared to the LPS condition.

**Figure 4 molecules-29-05623-f004:**
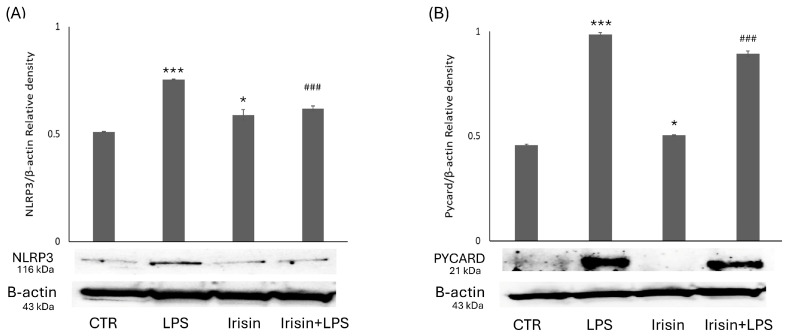
Evaluation of NLRP3 and Pycard expression following the administration of irisin, with or without LPSs. Western blotting detection and densitometric analysis of the expression of the pro-inflammatory NLRP3 (**A**) and Pycard (**B**) in control cells (CTR), BV2 cells treated with irisin, BV2 cells treated with LPSs (LPS), and BV2 cells treated with irisin + LPSs. The protein expression values are expressed in arbitrary units after normalization against β-actin. The data are presented as means ± SDs (* *p* < 0.05 vs. CTR; *** *p* < 0.001 compared to the control; and ### *p* < 0.001 compared to the condition of LPS-stimulated microglia).

**Figure 5 molecules-29-05623-f005:**
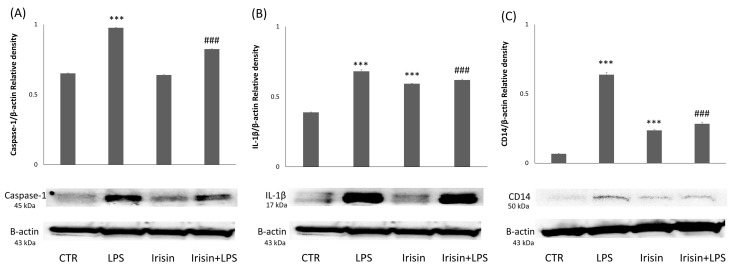
Evaluation of caspase-1, IL1Beta, and CD-14 expression following the administration of irisin, with or without LPSs. Western blotting detection and densitometric analysis of the expression of pro-inflammatory caspase-1 (**A**), IL1B (**B**), and CD-14 (**C**) in control cells (CTR), BV2 cells treated with irisin, BV2 cells treated with LPSs (LPS), and BV2 cells treated with irisin + LPSs (*** *p* < 0.001 compared to the control condition; and ### *p* < 0.001 compared to the condition of LPS-stimulated cells).

**Figure 6 molecules-29-05623-f006:**
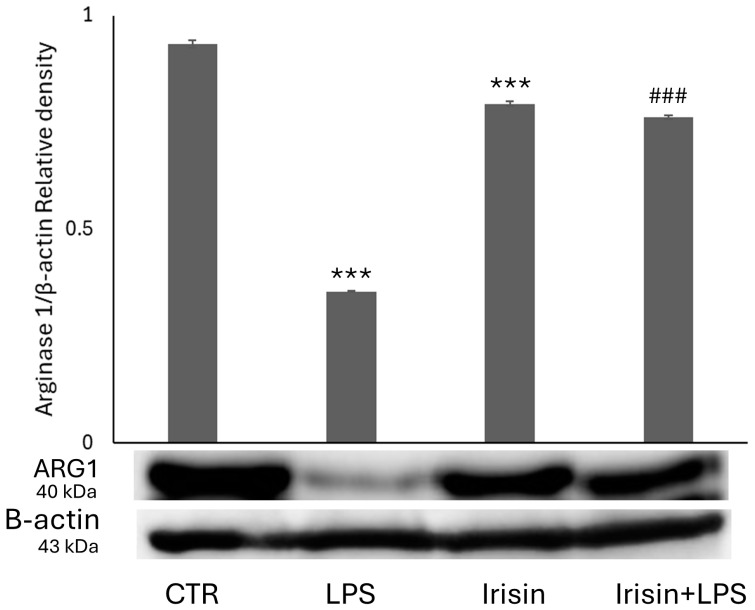
Evaluation of Arginase 1 expression following the administration of irisin, with or without LPSs. Western blotting detection and densitometric analysis of the expression of the anti-inflammatory agent in control cells (CTR), BV2 cells treated with LPSs (LPS), BV2 cells treated with irisin, and BV2 cells treated with irisin + LPSs (*** *p* < 0.001 compared to the control condition; and ### *p* < 0.001 compared to the condition of LPS-stimulated cells).

## Data Availability

The original contributions presented in the study are included in the article. Further inquiries can be directed to the corresponding author.

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
