# Peer review of "Irisin Attenuates Neuroinflammation Targeting the NLRP3 Inflammasome"

_molecules, 2024, doi:10.3390/molecules29235623_

Round 1
Reviewer 1 Report
Comments and Suggestions for Authors
1) The quality of the figures needs to be improved, the denominations of the axes are not distinguishable.
2) The cells in Figure 2 should be shown with a larger objective (e.g. 40X) to be able to differentiate the morphological changes of the cells. Part E in Figure 2 cannot be distinguished
3) Again, Figure 3 is unclear and the bar graph is not distinguishable. The relevant information in relation to the neuroprotective effect that it is intended to demonstrate is not clear.
4) Figures 4 and 5 cannot be distinguished.
5) Why is the inflammatory state induced with LPS, not β-amyloid?
6) The work should be completed by measuring other anti-inflammatory markers such as interleukins associated with anti-inflammation.
Author Response
"Please see the attachment."

Reviewer 2 Report
Comments and Suggestions for Authors
In the research " Irisin attenuates neuroinflammation target - ING NLRP3 inflammasome", Francesca Martina Filannino et al have looked at the role of Irisin - an anti-inflammatory peptide whose release is induced by exercise in tilting the balance towards a proinflammatory side and as a potential therapeutic strategy for neurodegenerative disorders. The overall idea of the research is good and remotely touches on how exercise is important but the research methodology itself is not novel. Overall, the results support the conclusions. A few revisions are needed as shown below.
1. The scale bars are missing in the images.
2. The brightness-contrast of all the images does not appear to be consistent.
3. The role of migration assay is not clear, needs to be clarified.
4. The magnification appears to be off, fig2 looks magnified compared to fig 1, again without scale bars, it is hard to compare.
Author Response
"Please see the attachment."

Reviewer 3 Report
Comments and Suggestions for Authors
The study provides valuable insights into irisin's neuroprotective and anti-inflammatory potential in modulating inflammatory pathways in BV2 microglial cells, particularly through the NLRP3 inflammasome pathway. However, there are some limitations in the study.
1. Only one concentration of LPS (1 μg/mL) and a narrow concentration range for irisin (5–20 nM) were used. Is there a rationale for using this? This may limit the dose-dependent effects and its effects at higher or lower concentrations.
2. The observations are based on a 24-hour treatment period. Is this a sufficient time period to understand the effects of irisin? This short timeframe may not provide insights into the long-term effects or sustained neuroprotective benefits of irisin, which are critical in the context of chronic neurodegenerative diseases.
3. What is the molecular mechanisms by which irisin modulates this pathway. A deeper mechanistic understanding in the discussion would clarify how irisin specifically interacts with the NLRP3 inflammasome.
Author Response
"Please see the attachment."

Round 2
Reviewer 1 Report
Comments and Suggestions for Authors
This work aims to evaluate the anti-inflammatory capacity of irisin in BV2 microglial cells by determining the down-regulation of markers such as NLRP3, Pycard, caspase-1, and IL-1beta and the migratory capacity of these cells without LPS stimulation and with or without irisin treatment. Cell morphology under different environmental conditions was also evaluated by optical microscopy.
The revised version of the work added a marker of M2 microglial polarization by determining the expression of Arg-1. The results obtained should be included in the manuscript abstract.
Although the quality of the figures has improved, the legends of the axes of the graphs are still difficult to read.
Author Response
"Please see the attachment."
